# An Insight to Nanoliposomes as Smart Radiopharmaceutical Delivery Tools for Imaging Atherosclerotic Plaques: Positron Emission Tomography Applications

**DOI:** 10.3390/pharmaceutics17020240

**Published:** 2025-02-12

**Authors:** Reabetswe Sebatana, Kahwenga D. Kudzai, Allan Magura, Amanda Mdlophane, Jan Rijn Zeevaart, Mike Sathekge, Maryke Kahts, Sipho Mdanda, Bwalya Angel Witika

**Affiliations:** 1Department of Pharmaceutical Sciences, School of Pharmacy, Sefako Makgatho Health Sciences University, Pretoria 0208, South Africa; sebatanareabetswe16@gmail.com (R.S.); daisykudzai@yahoo.com (K.D.K.); allablaq@gmail.com (A.M.); maryke.kahts@smu.ac.za (M.K.); 2Department of Nuclear Medicine, Steve Biko Academic Hospital, University of Pretoria, Pretoria 0028, South Africa; amanda.mdlophane@sanumeri.co.za (A.M.); mike.sathekge@up.ac.za (M.S.); 3Nuclear Medicine Research Infrastructure (NuMeRI), Steve Biko Academic Hospital, Pretoria 0028, South Africa; janrijn.zeervaart@necsa.co.za; 4Radiochemistry, The South African Nuclear Energy Corporation (Necsa) SOC Ltd., Pelindaba 0240, South Africa

**Keywords:** atherosclerosis, nanoliposomes, diagnosis, radiopharmaceuticals, positron emission tomography, single photon emission computerised tomography

## Abstract

Atherosclerosis is a chronic progressive disease which is known to cause acute cardiovascular events as well as cerebrovascular events with high mortality. Unlike many other diseases, atherosclerosis is often diagnosed only after an acute or fatal event. At present, the clinical problems of atherosclerosis mainly involve the difficulty in confirming the plaques or identifying the stability of the plaques in the early phase. In recent years, the development of nanotechnology has come with various advantages including non-invasive imaging enhancement, which can be studied for the imaging of atherosclerosis. For targeted imaging and atherosclerosis treatment, nanoliposomes provide enhanced stability, drug administration, extended circulation, and less toxicity. This review discusses the current advances in the development of tailored liposomal nano-radiopharmaceutical-based techniques and their applications to atherosclerotic plaque diagnosis. This review further highlights liposomal nano-radiopharmaceutical localisation and biodistribution—key processes in the pathophysiology of atherosclerosis. Finally, this review discusses the direction and future of liposomal nano-radiopharmaceuticals as a potential clinical tool for the assessment and diagnosis of atherosclerotic plaque.

## 1. Introduction

An estimated 17.9 million people died from cardiovascular disease (CVD) in 2019, accounting for 32% of all deaths worldwide, with 85% of these deaths due to heart attacks and strokes. CVDs are responsible for the increasing number of deaths worldwide, with 50% of them in developed territories, such as the United States of America, Japan, and Europe (45%) [1,2]. Atherosclerosis is one of the leading causes of most fatal CVDs [3]. Atherosclerosis is characterised by a chronic inflammatory response combined with an imbalance in lipid metabolism, causing narrowing of the arterial walls, which contributes to most cardiovascular conditions [4].

There is an increase in prevalence of atherosclerosis in developing territories that is largely attributed to an increased number of people with co-morbidities such as hypertension, diabetes mellitus, high cholesterol, and obesity. Between 1990 and 2022, the risk factors for atherosclerosis identified in the Sub-Saharan African region were smoking (10%), high cholesterol (25%), hypertension (30%), obesity (in women 2–40% and in men 1–15%), and diabetes (7%) [5]. Hypertension is a major risk factor for atherosclerotic mortality, responsible for approximately 26% of deaths worldwide [6]. Among the various risk factors that contribute to atherosclerosis, hypertension has the greatest impact on cerebrovascular infarctions (18%) and myocardial infarctions (48%) [7].

The most common strategy in the treatment or management of atherosclerosis is prophylactic drug therapy. On a clinical level, treating atherosclerosis is primarily focused on relieving CVD symptoms and preventing future cardiovascular events [8]. This can be attributed to the fact that the existing clinical screening methods for the diagnosis of atherosclerosis do not provide adequate information on the possible prognosis of the disease state before the first clinical presentation [8]. Plaque burden can be determined by plaque size, histology, chemical composition, and bioactivity [9]. Early detection presents new opportunities for primary prevention through lifestyle changes or even drug treatment, especially in patients with high cardiovascular risk [9].

Modern clinical imaging platforms for the diagnosis of atherosclerotic plaque formation, such as X-ray angiography and computed tomography (CT), focus on the post-symptomatic detection of morphological changes in the arterial wall that affect lumen diameter but do not provide direct information on plaque susceptibility [10]. Therefore, molecular imaging strategies have been developed for non-invasive diagnosis of coronary artery disease at various stages [11]. Nuclear medicine imaging methods can produce anatomical and functional images of various metabolic processes in the human body, which is a promising method for studying plaque formation, vulnerability, and rupture processes [12]. Recent advances in the understanding of the metabolic physiological processes that lead to atherosclerosis plaque formation have initiated the need for imaging techniques that provide sufficient information [13].

Single-photon emission computed tomography (SPECT) and positron emission tomography (PET) have the unique ability to target specific biological processes, such as inflammation, and can be combined with CT to measure volume and visualise patterns [14]. PET is particularly well suited for this purpose of imaging specific biological targets due to its non-invasiveness, high tissue penetration, and utilisation of high-sensitivity radiopharmaceuticals [15]. SPECT ligands have been prepared and investigated in animals to study different processes of atherosclerosis progression and rupture, including chemotaxis, angiogenesis, lipoprotein accumulation, proteolysis, and thrombus formation [16]. PET tracers such as fluorine-18-fluorodeoxyglucose ([^18^F]FDG), ^18^F-translocator protein ligand ([^18^F]TSPO), and ^18^F-choline ligand [16] have been studied in humans and have been shown to possess several advantages over SPECT, e.g., higher spatial resolution and quantitative evaluation of images [12].

Liposomes are nontoxic and biodegradable spherical biocompatible vesicles, consisting of concentric phospholipids, enclosed in an aqueous core [17]. Liposomes have proven potential use in both diagnosis and treatment of cardiovascular diseases, including atherosclerosis. The use of liposomes in atherosclerosis disease depends on their capacity to target either advanced plaques or early atherosclerotic lesions. Integrating liposomes with conventional diagnostic imaging tools can improve the detection of plaques by providing signals directly from the lesion site, laying out the morphology of the plaque [18]. The use of liposomes can be made multifunctional by the simultaneous loading of multiple contrast agents, which improves radiological image resolution [19].

This review article assesses the advancements in the development of tailored liposomal nano-radiopharmaceutical-based delivery tools and their use in atherosclerotic plaque diagnostics. This review also emphasises localisation and biodistribution of liposomal nano-radiopharmaceuticals as crucial mechanisms in the pathophysiology of atherosclerosis. Finally, the direction and future of liposomal nano-radiopharmaceuticals as a clinical tool for assessing and diagnosing atherosclerosis is analysed.

## 2. Pathophysiology and Aetiology of Atherosclerosis

The chronic nature of the disease, the suddenness of end-stage vascular events, the high frequency of atherosclerotic lesions in the arteries of asymptomatic individuals, and the absence of a single causative factor all made early detection and diagnosis challenging [20]. Cardiovascular diseases are multifactorial diseases, which involve genetic, environmental, and lifestyle factors. Traditional genetic risk factors for the disease include family history, age, sex, diabetes, hypertension, dyslipidemia, and obesity [21].

The risk of developing CVD increases with age and is generally higher in men, although postmenopausal women are also at increased risk [22]. Studies have indicated that sex hormones have powerful effects on the cardiovascular system, and hormonal dysfunction is associated with an increased risk of atherosclerosis [23,24,25]. Environmental and lifestyle factors such as smoking, diet, lack of exercise, alcohol consumption, infections, and air pollution strongly influence the development of the disease. Smoking is one of the leading causes of CVD, killing approximately 6 million people per year [26]. Smoking at a younger age (<50 years) has been found to be more harmful for women than for men, with the same number of cigarettes smoked per day increasing the harm [22]. Risk of stroke associated with increased BMI may be higher in men than in women [27].

Advances in understanding the cellular biology driving clinical events of atherosclerosis have highlighted the need for imaging techniques capable of providing information on the atherosclerotic plaque [28]. The need to understand the physiological changes that lead to the formation and eventual rupture of the plaque assists in the development of more specific imaging techniques. Each physiological change can be used to provide information on the prognosis of the disease using nuclear medicine imaging modalities. Atherosclerosis is a condition where lipids, fibrous materials, and calcium build up in the large arteries, causing them to narrow [29]. This process begins with activation of the endothelium, which leads to a chain of events that ultimately results in the formation of plaque in the arteries. As a result, cardiovascular complications occur, which are the leading cause of death globally [29,30].

### 2.1. Endothelial Dysfunction

The endothelium is responsible for maintaining the balance between dilation and constriction of blood vessels, preventing blood clotting, and promoting blood clot breakdown [31,32]. When this balance is disrupted, it leads to endothelial dysfunction, which leads to increased permeability, clot formation, and inflammation, all of which contribute to atherosclerosis. One of the earliest signs of atherosclerosis may be impaired vasodilation due to decreased production or activity of nitric oxide (NO) [33,34]. NO opposes the effects of vasoconstrictors and prevents platelet adherence, leukocyte infiltration, and smooth muscle cell proliferation, which leads to vasodilation [29,35]. It also prevents the oxidative modification of low-density-lipoprotein (LDL) cholesterol, which is thought to be a major cause of atherosclerosis [36]. Conversely, a lack of NO can lead to vasoconstriction, platelet aggregation, smooth muscle cell proliferation, leukocyte adhesion, and oxidative stress. Oxidised LDL cholesterol increases the production of caveolin-1, which inhibits the production of NO, and oxidative stress can also interfere with NO production through other mechanisms [29].

Haemodynamic forces in blood vessels can also lead to endothelial cell dysfunction [37]. Regions of the vessel where the flow is disturbed are particularly prone to developing lesions, as turbulent flow results in lower shear stress and a higher oscillatory index (Figure 1) [29]. This type of flow can also facilitate the infiltration of lipoproteins into the vessel intima by disrupting endothelial integrity and prolonging their residence in the area. The mechanical stimulus of blood flow also regulates the expression of various genes in endothelial cells, which further contributes to atherosclerosis [29].

### 2.2. LDL Infiltration

The accumulation of LDL in the bloodstream promotes its accumulation into the vessel intima. While it was previously believed that LDL crossed the endothelium by diffusion or paracellularly, it is now understood that transcytosis also plays a significant role in the transport of LDLs [38]. This process is mediated by scavenger receptor B1 (SR-B1) and activin A receptor-like type 1 (ALK1) receptors, which are co-localised with caveolae. Absence of caveolin-1, the major structural protein of caveolae in endothelial cells, significantly impairs LDL transport and retention within the arterial wall [29]. While further research is required to fully understand this process, these findings suggest that caveolae-dependent LDL uptake is an important factor in LDL transcytosis [29]. It is important to note that other factors, such as the glycocalyx, pericytes, the subendothelial extracellular matrix, and shear stress, also play a role in LDL infiltration. LDL particles get trapped in the subendothelial space, where they undergo oxidation due to a lack of protective antioxidants [39]. Oxidised LDLs contain oxidised lipids and promote atherosclerotic plaque development. LDLs can be oxidised by free radicals in the extracellular media or directly by enzymatic activity, which is followed by a small degradation of polyunsaturated fatty acids, leading to the formation of conjugated dienes and short-chain aldehydes [39]. Depending on the LDL oxidation level, oxidised LDLs are classified as minimally modified (Mm) LDL or extensively oxidised LDL. Mm-LDLs induce a pro-inflammatory response in endothelial cells and macrophages, while extensively modified oxidised LDLs are recognised by a range of scavenger receptors, allowing non-regulated uptake by scavenger receptors [29].

### 2.3. Endothelial Activation

Endothelial stimulation, also known as endothelial type I stimulation, refers to the response of the endothelium to inflammatory agents that cause changes in microvascular tone, permeability, or leukocyte diapedesis [29]. This response is acute and results in short-term functional and morphological changes without requiring de novo protein synthesis or gene upregulation. However, certain proinflammatory agents can lead to sustained phenotypic modulation, known as endothelium type II activation, resulting in a complex inflammatory response. This activation is initiated by increased NF-kB production within the endothelial cells and leads to the upregulation of leukocyte adhesion molecules, chemokines, and prothrombotic mediators [29].

Monocyte recruitment and foam cell formation are also facilitated by activated endothelial cells, which induce selective monocyte recruitment into the intima [40]. Monocyte recruitment involves monocyte capture, rolling, adhesion, activation, and transmigration into the intima space [41]. Chemokines secreted in response to proinflammatory signals facilitate this process, with MCP-1 being the most frequent chemokine mediating monocyte transmigration. Once in the intima, monocytes differentiate into macrophages that can be polarised into the M1 or M2 phenotype, depending on the inflammatory environment. Macrophages express receptors that mediate the internalisation of modified and non-modified LDLs [29].

### 2.4. Fibrous Plaque

A fibrous plaque develops, and atheroma plaques go through a phase where they transform from a fatty streak to intimal growth. During this process, a region called the necrotic core emerges, which is a cell-free and lipid-rich area [42]. This stage is identified by the presence of a necrotic core, and to make the plaque stable, it is covered with fibres, resulting in the development of a fibrous cap. The combination of the necrotic core and fibrous cap is recognised as the characteristic feature of advanced atherosclerosis, and it is improbable for atheroma plaque to regress at this point [29].

### 2.5. Plaque Calcification

In this stage, the atheroma plaque calcification takes place. This calcification process is initiated in regions with reduced collagen fibres due to inflammation. The death of macrophages and synthetic vascular smooth muscle cells (VSMCs) results in the release of matrix vesicles, which triggers the calcification process of the plaque [32]. Accumulation of calcium orthophosphate occurs at the nucleation sites, which is eventually converted into amorphous calcium phosphate and then into crystalline structures. Other factors, such as reduced levels of mineralisation inhibitors or increased osteogenic trans differentiation, also contribute to the calcification process [32,35].

Notably, during atherosclerosis development, pericytes and VSMCs trans-differentiate into osteoblast-like phenotypes, acquiring the capacity to generate a mineralised matrix that leads to the formation of calcium deposits, as seen in bone tissue. This process leads to microcalcifications, which is the early stage of the vascular calcification cascade in both the intima and media.

Larger calcifications that reach from the bottom of the necrotic core to the surrounding matrix are subsequently formed from these microcalcifications [30]. The quantity and size of calcium deposits may not always correspond with plaque vulnerability, even though calcification is a sign of advanced atherosclerosis and is positively correlated with plaque size. Instead, other features, such as location, calcification type, or the surrounding environment, may be more closely linked to plaque vulnerability [29,32].

### 2.6. Plaque Rupture and Thrombus Formation

Low shear stress in branched areas contributes to the development of atheroma plaques, which initially prevent lumen narrowing through outward vessel remodelling. However, this prolongs low wall shear stress conditions and aggravates plaque growth, making it more rupture prone [29]. Plaque vulnerability is characterised by a large necrotic core, a thin fibrous cap, and an increased inflammatory response [43]. This inflammation promotes the instability of the fibrous cap, making it susceptible to rupture when exposed to haemodynamic forces. When the plaque ruptures, platelets adhere and become activated, leading to the formation of a thrombus that covers the lesion, promoting the expansion of the intima to the luminal side [29]. The cascade that leads to an atherosclerotic plaque can be summarised in six phases, as illustrated in Figure 2.

In summary, atherosclerosis is a progressive arterial wall chronic inflammatory disease which remains asymptomatic until the rupture of plaque, leading to acute atherothrombotic events [4]. The three cellular components of the circulation, i.e., monocytes, T lymphocytes, and platelets, as well as two cell lines of the arterial wall, i.e., endothelial and smooth muscle, interact in many ways to produce lesions of atherosclerotic injury [44]. Plaque formation begins when small cholesterol crystals deposit in the underlying smooth muscle and endothelium [44]. The formation of bulges inside the artery is a result of the plaque growing with the proliferation of the surrounding fibrous and smooth muscle tissue, which subsequently reduces blood flow [44]. The generation of connective tissue by fibroblasts and the deposition of calcium in the lesion causes hardening of the arteries [44]. The uneven surface of the artery leads to the formation of a thrombus, resulting in a sudden blockage of blood flow.

## 3. Imaging Atherosclerosis

The current imaging methods can be categorised as invasive (coronary angiography, intravascular ultrasound, and optical coherence tomography) and non-invasive (ultrasound, CT, magnetic resonance imaging (MRI), and scintigraphy techniques). Each technique has the limitation of only imaging certain physiological changes during atherosclerotic plaque formation, or it only images the atherosclerosis plaque but cannot determine if it will rupture. According to Ibañez and colleagues [9], carotid ultrasound provides a measure of intima media thickness (IMT), which is associated with cardiovascular events and risk factors.

MRI is a functional and anatomical imaging technique that has low spatial resolution but offers excellent soft tissue contrast. MRI can reveal the total amount of plaque on different vascular beds, although resolution is limited to large vessels. MRI is the guiding principle for carotid artery plaque characterisation and is best able to differentiate between “soft” plaque components, such as lipid material, and haemorrhages. Although it has the advantage of being non-irradiating, the acquisition process is time consuming, and operator expertise is required to accurately interpret the acquired images [45].

Excellent sensitivity, high spatiotemporal resolution, and real-time imaging are features of fluorescence imaging technology that make it ideal for direct observation of atherosclerotic molecular processes and abnormalities [46]. Fluorescence imaging can be employed to visualise the accumulation of plaque in blood vessels and monitor disease progression. By targeting specific molecular markers of atherosclerosis, such as inflammatory cells or lipid deposits, fluorescent dyes can be used for specific labelling and visualisation of these disease-related processes [47]. The absence of clinically licensed tracers that target aspects of plaque biology is one of the current constraints of fluorescence imaging. This is partly because of the low sensitivity to molecular imaging agents, the possibility of toxicity from systemic injection, the expense, labelling, and regulatory concerns.

CT scans can be used to determine the load and type of coronary plaque, unlike MRI and ultrasound, which require ionising radiation. Molecular imaging by MRI and/or PET or SPECT may reveal inflammatory activity in atherosclerotic plaques. An imaging modality capable of non-invasively quantifying the atherosclerotic plaque component, identifying vulnerable atherosclerotic lesions for rupture, and monitoring the impact of both surgical and drug interventions will improve the clinical outcome of cardiovascular related diseases [48]. However, the anatomic structure of cells suffers from restricted visualisation due to low spatial resolution when utilising SPECT or PET imaging.

In the last century, there has been a growing interest in nanotechnology and its possible application in health [49]. Nanotechnology is the science and engineering concerned with the design, synthesis, characterisation, and application of nanoscale materials and devices [50]. Nanomedicine has the potential to be used for therapy, cancer targeted therapy, drug and gene delivery, tissue engineering, and imaging [51].

Radiolabelling of nanomaterials has been performed using different radionuclides, with imaging radionuclides such as technetium-99m (^99m^Tc) and copper-64 (^64^Cu), and therapeutic radionuclides such as lutetium-177 (^177^Lu) and radium-223 (^223^Ra). In addition, radionuclides such as carbon-14 (^14^C) [52], gallium-68 (^68^Ga) [53], zirconium-89 (^89^Zr) [8], iodine-125 (^125^I) [54], yttrium-90 (^90^Y) [55], gold-199 (^199^Au) [56], barium-131 (^131^Ba) [57], etc., have also been used for radiolabelling of nanomaterials in radiopharmaceuticals.

### 3.1. Radionuclides

Atherosclerotic imaging technology focused almost exclusively on identifying anatomical obstructions of outflow until the recent understanding of the pathophysiological pathway of the disease [28,58]. Finding coronary artery susceptible plaques has become more crucial in order to help identify patients who are at high risk for cardiovascular disease [59]. Radiopharmaceuticals in PET and SPECT are typically used in conjunction with other imaging modalities, such as CT or MRI, to provide a comprehensive evaluation of atherosclerosis disease progression [60].

Currently, the most utilised radionuclides for the imaging of atherosclerosis are technetium-99m (^99m^Tc), copper-64 (^64^Cu), indium-111 (^111^In), fluorine-18 (^18^F), and gallium-68 (^68^Ga) [61]. These radionuclides provide 2- and 3-dimensional (2D and 3D) reconstructed tomographic images that supply a functional representation of the atherosclerotic plaque by utilising the pathophysiological changes caused by the disease [58]. Summary of the most commonly used radionuclides for scintigraphy imaging of artherosclerosis is provided in Table 1.

#### 3.1.1. SPECT Radiopharmaceuticals

SPECT design is based on linear motion of the transducer systems at each projection angle producing 3D images of the distribution of radiotracers injected into the bloodstream and providing detailed physiological information about the tissue [63]. Technetium-99m (^99m^Tc) is the most commonly used radionuclide in SPECT imaging [59]. It is produced by a molybdenum-99/technetium-99m generator, has a half-life of 6 h, and emits 140 keV gamma rays during radioactive decay [64].

#### 3.1.2. PET Radiopharmaceuticals

PET imaging is believed to have high sensitivity and more specific radioligands to target the metabolic processes involved in atherogenesis and plaque disruption, such as inflammation, where radioligands target macrophages (e.g., [^18^F]FDG), microcalcification (e.g., ^18^F-sodium fluoride ([^18^F]NaF)), and hypoxia (^18^F-fluoromisonidazole ([^18^F]FMISO)) [59]. PET imaging uses positron-emitting radionuclides, such as ^68^Ga, ^11^C, and ^18^F, to detect the different biochemical and metabolic substrates by using coincidence detection of two 511 keV photons emitted in opposite directions after annihilation [65] (Saha, 2010). PET offers a 2- to 3-fold increased spatial resolution when compared to SPECT [66].

Nuclear medicine and molecular imaging provide markers of plaque vulnerability, including inflammation and neovascularisation. PET combined with CT, i.e., PET/CT, is a hybrid imaging modality that provides high sensitivity from the PET component combined with anatomic detail of the atherosclerotic vascular territories from the CT component. On the other hand, it still faces the challenge of resolution for the imaging of small vessels such as coronary arteries [67].

#### 3.1.3. Nanoliposome Technology

Spherical bi-layered lipid vesicles known as liposomes are characterised as having either a unilamellar or multilamellar phospholipid bilayer. The phospholipid components assemble and aggregate spontaneously in a cluster to form an enclosed amphiphilic system with an aqueous centre where the polar heads are facing outward and the hydrophobic tails are facing inward [68,69], as depicted in Figure 3.

Biological components like proteins, lipids, and nucleic acids, as well as therapeutic drugs, immunomodulators, and radionuclides, can all be delivered spatiotemporally by liposomes [70]. There are different strategies used to radiolabel the radionuclide to the liposomes, which include surface coupling and inner incorporation. Surface coupling is the anchoring of radionuclides on the surface of nanomaterials using either indirect surface labelling or direct surface labelling. Direct surface radiolabelling is a method in which there is a physical interaction between nanoparticles (NPs) and radionuclides [71]. When radiolabelling liposomes using surface coupling, the most used approach is indirect surface labelling using chelators, which act as a bridge between NPs and radionuclides. When the NPs are produced with the radionuclide infused into the nanomaterial, this is referred to as inner incorporation. The most commonly used inner incorporation for liposomes is encapsulation, where the radionuclide is internalised into the cavity and gallery of nanomaterials [72]. A summary of the labelling types and techniques is depicted in Figure 4.

Liposomes are biocompatible and biodegradable vesicles that can encapsulate radionuclides. The outside layers of the lipid layer serve as a boundary, ensuring the bioactive payload from chemical breakdown, corruption, or protein binding. Prior to radiolabelling, the choice of radionuclide and liposomal definition ought to be taken into account based on the investigate purposes and the required applications of the radio-liposomes. The half-life of radionuclides ought to be congruous with the natural half-life of the liposomes [70].

The half-life of liposomes in the circulatory system is determined by the size and lipid composition of liposomes in the system.

A vital component for liposome longevity, the mononuclear phagocytic system (MPS), often referred to as the reticuloendothelial system (RES), is regulated by macrophages and phagocytic cells in the liver, spleen, and bone marrow. Splenic macrophages remove liposomes larger than 200 nm, while liver Kupfer cells remove liposomes that are smaller than 70 nm due to the limited pore size of liver sinusoidal filters [70]. Phagocytic cells catalyse liposomes by recognising plasma and blood cells absorbed on the surface of liposomes.

To evade the RES with the aim of prolonged circulatory longevity, manipulation of liposomes is mandatory as well as the prevention of the premature release of active materials which in turn will increase the delivery and the accumulation of active materials at desired sites [74]. Liposomes have been used recently in clinical trials due to their long-term safety, with the long-term application of liposomal nano-formulations being well tolerated in cancer patients [75]. The disadvantage to synthetic preparation of liposomes is that they tend to be suspectable to premature release of active material before delivery to the target site. This may be due to the fact that natural phospholipids with unsaturated fatty acid groups, which are prone to breakdown, are typically used to create traditional liposomes [74].

Liposome surfaces are more hydrophilic. Coating them with hydrophilic substances like polymer polyethylene glycol (PEG), silica acid, or monosialoganglioside (GM1) can reduce the amount of protein opsonins that liposomes can adsorb and increase vascular permeability. This leads to an increased accumulation of liposomes in tumours. A number of liposomal formulations have received clinical approval, including the first FDA-approved liposome formulation with a size of less than 100 nm for treating ovarian cancer and multiple myeloma, pegylated liposomal doxorubicin (DOX); daunorubicin-encapsulated liposome for Kaposi’s sarcoma; a cytarabine-encapsulated liposome for lymphomatous meningitis; and the most recent vincristine-encapsulated liposome for acute lymphoblastic leukaemia. Numerous other liposomal formulations are still undergoing preclinical or clinical studies [76].

By combining the diagnosis of atherosclerosis with nanoliposomes, biodistribution can be significantly enhanced, leading to better clinical information on the severity of the disease, better uptake into the plaque, and improved blood clearance [77]. Research has demonstrated that long circulating nanoliposomes radiolabelled with long-lived radionuclides can be injected intravenously and accumulate in atherosclerotic lesions in both cardiovascular disease humans and animal models with increased permeability. PET imaging of radiolabelled nanoliposomes can nonetheless offer a quantitative assessment of biodistribution in vivo, even though its limited spatial resolution makes it difficult to see detailed plaque formation [77].

Liposomes have demonstrated quick blood clearance and strong macrophage absorption. Rapid blood clearance lowers the high background caused by elevated blood radioactivity for atherosclerotic plaque diagnostic imaging [78]. According to Lamichhane et al. (2018) [79], using radiolabelled nanoparticles for molecular imaging also aims to improve delivery, monitoring in vivo pharmacokinetics, and enable well-controlled release. High integration efficiency and adequate retention of radiolabelled drugs are necessary for the use of radiolabelled liposomes in clinical settings. These liposomal compositions can improve uptake or bypass the drawbacks of traditional treatments. Because of their adaptability in surface functionalisation, nanoparticles offer chances to improve target specificity and label them with different isotopes, which makes them useful as contrast agents. Several labelling techniques can be used to radiolabel liposomes, although doing so may change the pharmacokinetics of the radiolabelled radiopharmaceutical liposome compared to radiopharmaceutical [79].

#### 3.1.4. Targeting Liposomes to Atherosclerosis

Liposomes’ capacity to target early atherosclerosis lesions or advanced plaques is largely responsible for the efficient transport of medications, genes, cells, and contrast agents in atherosclerotic disease. PEGylation is a technique for altering liposomes [19]. The enhanced permeability and retention (EPR) effect is the passive accumulation and detention of liposomes at aberrant tissues [79]. PEGylation lengthens the duration of liposome circulation, increasing their ability to target plaques. Numerous cutting-edge methods for liposome targeting of plaques have been studied. These tactics cover the several phases of atherosclerosis progression, focusing on early endothelial dysfunction, macrophage lipid accumulation, and, finally, susceptible plaques that are about to rupture, which is a feature of advanced atherosclerotic disease [18].

The two targeting techniques linked to drug delivery via nanoparticles are passive and active targeting. The prepared drug carrier complex circulates through the bloodstream and is driven to the target site by affinity or binding influenced by properties like pH, temperature, molecular site, and shape of the NP to determine how much is accumulated in the region of the arterial wall in the atherosclerotic lesion [80]. Passive targeting is nonspecific and depends on the EPR effect. Liposomes must be coupled with functional group molecules such as proteins, antibodies, antibody fragments, carbohydrates, and other small molecules in order to offer a high level of specificity. The ability of liposomes to identify cells that express corresponding receptors is essential following surface changes, and this process is known as active transport [18].

In atherosclerosis, nanomaterials can be delivered to multiple target areas, including the extracellular matrix, inflammatory cells, cell adhesion molecules, and proteases. Drug solubility in drug delivery frequently varies from one drug to another because of various characteristics, such as hydrophilic and hydrophobic medicines. Hydrophilic medications have several problems, such as fast metabolism, renal clearance, and poor pharmacokinetics brought on by their incapacity to cross lipid membranes efficiently. Drugs that are hydrophobic experience less than ideal delivery because of their limited water solubility, which results in low bioavailability [80].

### 3.2. Radiolabelled Liposomes as Nuclear Imaging Probes for Atherosclerosis

#### 3.2.1. Preclinical Evidence

Recently, the use of nanoparticulates has emerged as an experimental treatment option for atherosclerosis [77]. By boosting the targeted accumulation of small-molecule medications into sick tissue while minimising systemic toxicity, nanomedicine-based drug delivery seeks to enhance the treatment of disease. Liposomes must be coupled with functional group molecules such as proteins, antibodies, antibody fragments, carbohydrates, and other small molecules to offer a high level of specificity. To enable active transport following surface modifications, liposomes must be able to identify cells that express appropriate receptors [70].

Nanoliposomes may accumulate in plaque macrophages through two different ways. They can first enter the plaque through circulating monocytes or the spleen, or they can extravasate because of the blood vessel wall’s increased permeability. This causes long-circulating nanoparticles to gather in the subendothelial space before being phagocytosed by plaque macrophages [77]. The element ^89^Zr has been investigated as a PET radionuclide for the imaging of metabolic atherosclerotic plaques. The long physical half-life of ^89^Zr (78.4 h) allows the monitoring of long-circulating materials where blood pool signal clearance is required, e.g., blood vessel wall targeting. To assess biodistribution and blood vessel wall targeting, Lobatto et al. (2019) intravenously injected rabbits with a single dose of ^89^Zr-liposomes, subsequently subjecting them to serial in vivo PET/CT and PET/MRI sessions on clinical scanners [77]. Vascular permeability was measured ex vivo using near-infrared fluorescence (NIRF) imaging and in vivo using three-dimensional dynamic contrast-enhanced MRI (3D DCE-MRI). High uptake of ^89^Zr-liposomes was found in the abdominal aortic artery wall when the maximum standard uptake value (SUVmax) of the total vessel wall was measured in atherosclerotic rabbits.

The uptake in the atherosclerotic rabbits doubled after clearance from blood when compared to the control rabbits from day 3. Visualisation of radioactivity accumulation in the aorta of atherosclerotic rabbits was only obtained at day 3 post-administration. The long half-life of ^89^Zr allowed imaging and non-invasive monitoring of liposomes for up to 15 days post-injection [77].

One typical characteristic of atherosclerosis-vulnerable plaques is macrophage infiltration. Apoptotic cells are recognised by macrophages as having exposed phosphatidylserine (PS), which causes the macrophages to ingest the apoptotic cells through phagocytosis [78]. PS liposomes were prepared by lipid film hydration and modified to express PS100 and PS200 liposome. Phosphatidylcholine (PC) liposomes were prepared as controls and modified to express PC100 and PC200 liposomes. A remote loading method was used to encapsulate ^111^In-nitrilotriacetic acid using 100 or 200 nm liposomes.

After two hours of incubation with cultivated macrophages, the absorption level of the ^111^In-loaded liposomes was assessed. For biodistribution studies, ddY mice were used and for autoradiography the aortas of apolipoprotein E–deficient (apoE2/2) mice were harvested, followed by oil red O staining. Then, ^111^In-loaded liposomes were injected intravenously into Watanabe heritable hyperlipidaemic rabbits, followed by SPECT imaging 48 h after injection. The atherosclerotic region was successfully detected by ^111^In-PS200 in apoE2/2 mice and Watanabe heritable hyperlipidaemic rabbits in vivo. In order to improve the SPECT images, liposome modification would be required to slow down blood clearance and decrease liver uptake [81]. By using radiolabelled liposomes (PS liposomes) to target macrophages, the atherosclerotic region was identified; nevertheless, more research is required to enhance the in vivo biodistribution and plaque accumulation level [82].

Of all the liposomes examined, ^111^In-labelled 1,2-Distearoyl sn-glycero-3-phosphoglycerol (DSPG) liposomes exhibited the greatest absorption by RAW264 cells (Figure 4). Particularly, ^111^In-labelled DSPG liposomes are a biocompatible liposome component that has demonstrated quick blood clearance and targeted accumulation in organs with a high concentration of macrophages. Additionally, ApoE^−/−^ animals injected with ^111^In-labelled DSPG liposomes showed accumulation of macrophages in the plaques with a high target-to-nontarget ratio (TNR). These findings imply that DSPG-containing ^111^In-labelled liposomes are effective nuclear imaging probes for identifying atherosclerotic plaques [82]. A summary of this work is provided in Table 2.

#### 3.2.2. Clinical Evidence

The pathological progression of plaque formation has become the building block for researchers when developing NPs as delivery vehicles for therapeutic or imaging agents. Selective targeting of plaques for imaging and therapeutic purposes remains a challenge due to lack of efficient delivery of the assembled nanoparticles to specific tissues with cell specificity and subcellular precision. There are still safety concerns which need to be examined despite numerous preclinical studies on NPs exhibiting prolonged circulation and excellent vitality for managing atherosclerotic plaques, which highlights the gap between laboratory animal models and clinical patients [83]. Since radiopharmaceuticals are typically administered at low dosages, the therapeutic usage of radiolabelled nanomaterials can reduce their toxicity concerns. When compared to conventional carriers, gamma-emitting radionuclide-radiated nanomaterials can offer imaging probes useful extra characteristics [62] (Figure 5).

Ogawa et al. [84] exploited the presence of infiltrated macrophages in vulnerable plaque by radiolabelling PS liposomes with indium-111 to image atherosclerotic plaque. Due to the proximity with the liver, whole body images were more difficult to interpret, despite encouraging ex vivo autoradiography results. Therefore, clinical translation showed no significant reason for further investigation [84]. This is an intriguing use of radiolabelled liposomes as diagnostic tools, while it is likely that a different radionuclide would yield better results [84].

In a first-in-human small-scale clinical trial, ^89^Zr-labelled liposomes in the atherosclerotic vessel wall showed no clinical effect. Patient identification through screening procedures to identify subjects that would most likely benefit from therapy was shown to be an important factor. To optimise atherosclerosis NP therapy and diagnosis, tools to monitor targeting, atherosclerotic burden, and biodistribution should be identified in order to assist in patient selection [81].

Before clinical translation can be achieved using NPs, further research is needed for optimisation of imaging and therapy of atherosclerotic plaques. The mechanism of distribution within the microenvironment of the plaque, e.g., NP physiochemistry and surface functional ligands, needs to be further examined to assess their effect on the plaque [83].

## 4. Future Perspective

Clinical application and cutting-edge research in cardiovascular diseases have been advanced by nanoparticle development [85]. One area of interest has come from developing nanoparticle-based platforms for multimodal imaging and theranostic applications. The versatility offered by nanoparticles provided numerous advantages such as large surface area-to-volume ratio, easy surface modification, and the ability to incorporate various imaging agents and therapeutic payloads, despite their small size [86]. Liposomes can incorporate various imaging agents, such as fluorescent dyes, radionuclides, or contrast agents, for combined imaging. Liposomes can be functionalised with targeting ligands or stimuli-responsive moieties to be utilised in cardiovascular diseases [85].

The possibility of nanoparticle multimodality as an imaging probe for the imaging of atherosclerotic plaque has been researched by Nahrendorf et al. [69]. They developed a trimodality (PET, MRI and fluorescence) imaging probe utilising ^64^Cu radionuclide with a magnetic nanoparticle base material and a near-infrared fluorochrome. Studies have used MRI to assess areas of inflamed lesions in animals and in carotid artery plaques in patients. Nuclear techniques such as PET/CT provide higher sensitivity at lower concentration compared to MRI [69]. The trimodality highlights the interest in synergistic hybrid imaging systems which are currently being utilised in clinical settings to improve image sensitivity, patient radiation exposure, and diagnostic accuracy [69].

## 5. Conclusions

The use of liposomal nano-radiopharmaceuticals in PET imaging offers several advantages over conventional imaging techniques. Firstly, the liposomal formulation helps to protect the radioactive isotope from degradation, ensuring optimal image quality. Additionally, the small size of liposomes enables them to penetrate the plaque, providing a detailed assessment of the plaque composition and vulnerability.

Moreover, by incorporating targeting ligands on the surface of liposomes, such as antibodies or peptides that recognise specific markers associated with atherosclerosis, the imaging agent can be further enhanced for selective plaque imaging. This targeted approach improves the sensitivity and specificity of PET imaging, enabling the detection of early-stage plaques and potentially identifying high-risk plaques prone to rupture.

PET liposomal nano-radiopharmaceuticals have shown promising results in preclinical studies, demonstrating their ability to accurately visualise atherosclerotic plaques in animal models. These imaging agents hold great potential for clinical translation, as they can aid in the early detection, risk stratification, and monitoring of atherosclerosis, ultimately leading to improved patient management and personalised treatment strategies.

For future perspective, further research and clinical trials are needed to validate their efficacy and safety in human subjects. In order to expedite their clinical translation and address the existing challenges associated with NPs for imaging and therapy of atherosclerotic plaques, the following can be looked into: physicochemical interactions between nanoparticles and biological systems, systematic evaluation of biotoxicity and immunogenicity, and finally, an exploration and optimisation of NP size, surface potential, and morphology [82].

## Figures and Tables

**Figure 1 pharmaceutics-17-00240-f001:**
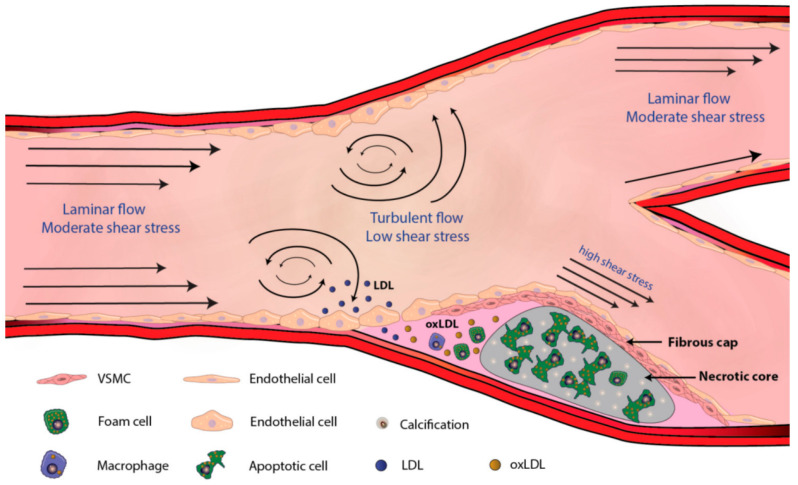
Effects of Haemodynamic forces on atherosclerotic plaque formation: The flow separation; haemodynamic force is present at branch locations. The initial stage of atheroma plaque formation; endothelial dysfunction and LDL infiltration are encouraged by disturbed laminar or turbulent flow, which also lowers wall shear stress [29].

**Figure 2 pharmaceutics-17-00240-f002:**
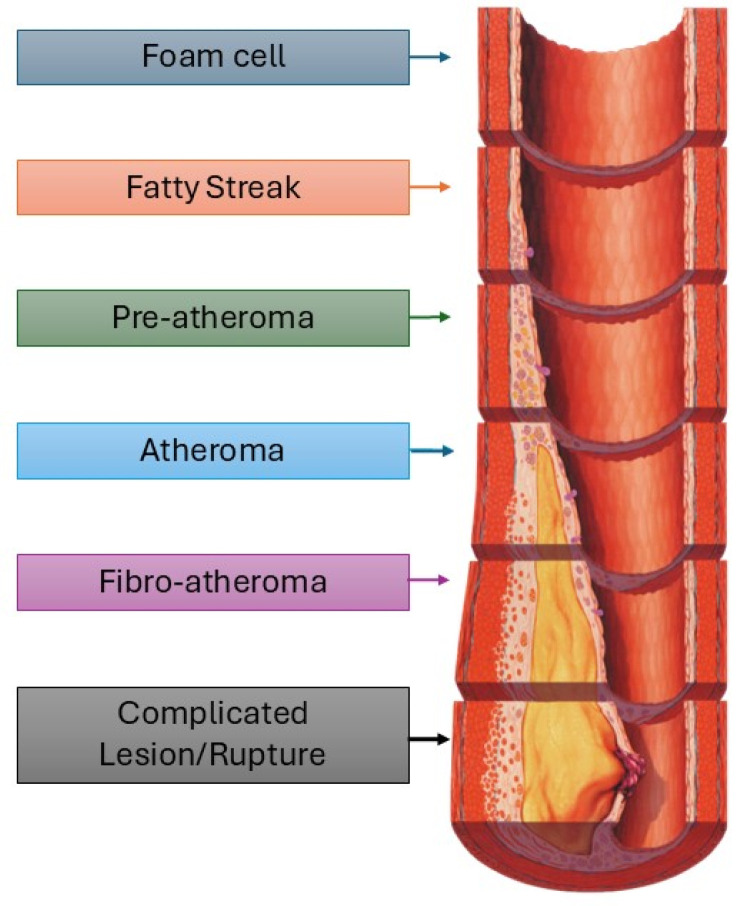
Formation of atherosclerotic plaques and the evolution of the plaques from fatty streak or atheroma to plaque rupture and finally to blocked and thrombosed artery.

**Figure 3 pharmaceutics-17-00240-f003:**
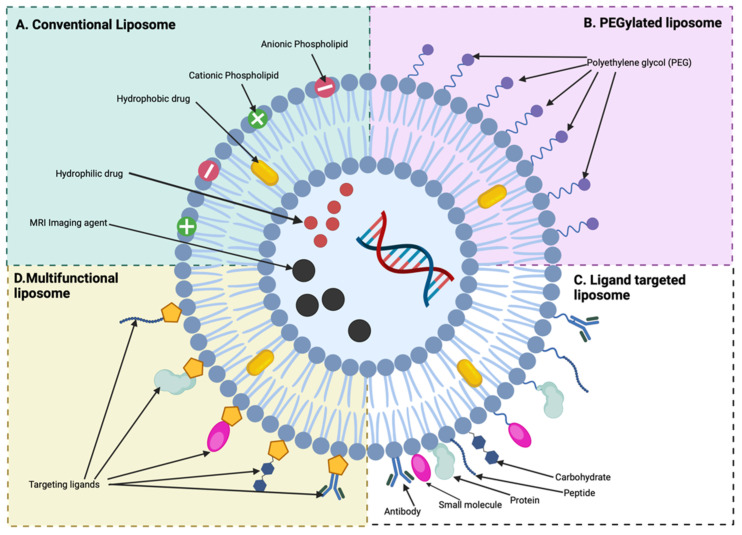
Structure of conventional and functionalised liposomes which can be used for diagnosis and treatment.

**Figure 4 pharmaceutics-17-00240-f004:**
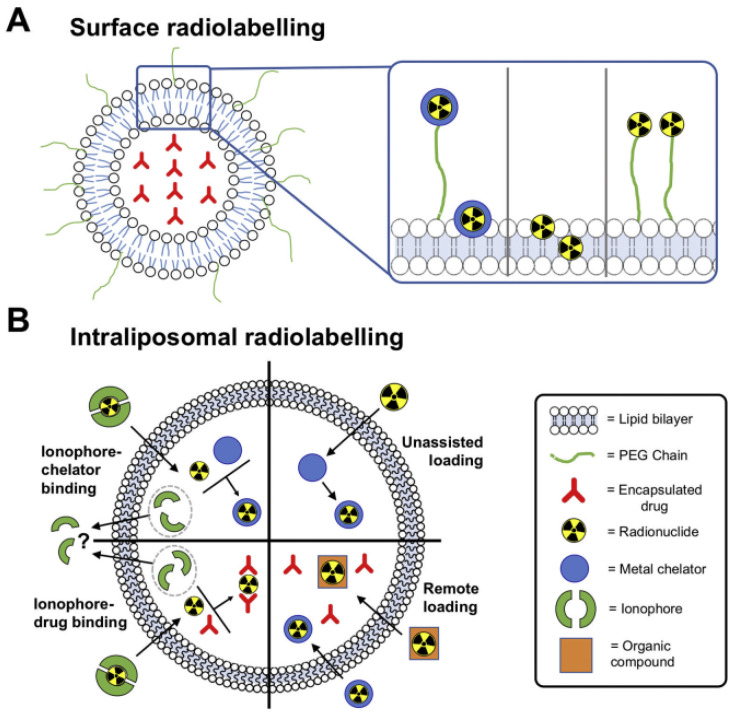
Diagrammatic representation of the various liposome radiolabelling techniques. (**A**) Surface radiolabelling: the radionuclide can be integrated directly into the lipid bilayer or connected to the liposomal membrane via a PEG chain, chelator or not. (**B**) Intraliposomal radiolabelling: the aqueous core encapsulates the radionuclide. Radioactive substances or complexes can passively cross the bilayer and become trapped, or ionophores can be employed to move radionuclides across the bilayer, where they can be bound by chelators or medications inside the liposomes [73].

**Figure 5 pharmaceutics-17-00240-f005:**
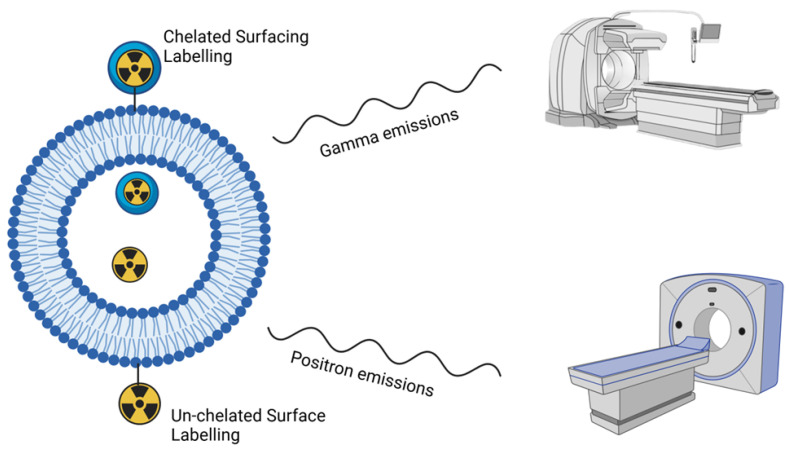
Schematic diagram, nanoparticles labelled with gamma-emitting and positron-emitting radioisotopes. Figure created in the Mind the Graph platform, available at www.mindthegraph.com (accessed on 25 January 2025).

**Table 1 pharmaceutics-17-00240-t001:** Most commonly utilised radionuclides for scintigraphy imaging of atherosclerosis [62].

Radionuclides	Emission Type	Emax (γ) [keV]	Half-Life
^67^Ga	Auger, γ	93, 184, 300, 393	78.3 h
^111^In	Auger, γ	171, 245	67.2 h
^99m^Tc	Positron	140	6.0 h
^18^F	Positron	511	1.83 h
^64^Cu	γ (15%), β	511	12.7 h

**Table 2 pharmaceutics-17-00240-t002:** Summary of the preclinical evidence with the mechanisms of radiolabelling and uptake and their limitation.

Radionuclide	Precursor	Mechanism of Radiolabelling	Mechanism of Uptake	Limitation	Ref
Zirconium-89 (78 h)	PEGylated Liposomes	Surface labelling	Macrophage infiltration Migration into plaque through splenic or circulating monocytes OR Extravasating due to increased permeability of the blood vessel wall.	No clinical effect in first-in-human small-scale clinical trial and exact cause could not be identified.	[77]
Indium-111 (67.3 h)	Phosphatidylserine (PS) PS100 liposome PS200 liposome Phosphatidylcholine (PC) PC100 liposome PC200 liposome	Remote loading was used to encapsulate the radionuclide.	Apoptotic cells expressing phospholipid are engulfed by macrophages through phagocytosis.	A slower blood clearance of PC liposomes than PS liposomes was observed, which can result in excessive liver uptake, making it difficult to visualise the small atherosclerotic region in coronary arteries.	[81]
Indium-111 (67.3 h)	1,2-Distearoyl sn-glycero-3-phosphoglycerol (DSPG) liposomes	Remote loading was used to encapsulate the radionuclide.	DSPG is analogue for phospholipid, which expressed on the surface of apoptotic cells is recognised and phagocytosed by macrophages.	Comparison between the DSPG liposome and the control DSPS liposome showed that DSPG liposomes have a faster blood clearance, which reduced the sensitivity and specificity of the images.	[82]

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
