# Peer review of "An Insight to Nanoliposomes as Smart Radiopharmaceutical Delivery Tools for Imaging Atherosclerotic Plaques: Positron Emission Tomography Applications"

_pharmaceutics, 2025, doi:10.3390/pharmaceutics17020240_

Round 1

Reviewer 1 Report

Comments and Suggestions for Authors

1. Please include a line on nanoliposomes in the abstract 

2. Include nanoliposomes in keywords instead of liposomes 

3. The author can summarise the etiology of atherosclerosis in tabular form

4.  Please include the advantages and disadvantages of different imaging techniques in detail

5. Inlude more relevant studies in the preclinical evidence

6. A summary image of all preclinical studies would improve the quality of the manuscript 

7. Inlude the table for preclinical studies for a better understanding along with  limitations 

Author Response

Reviewer 1

1. Please include a line on nanoliposomes in the abstract 

Tracked changes have been highlighted in red. Lines 21-23

2. Include nanoliposomes in keywords instead of liposomes 

Tracked changes have been highlighted in red.

3. The author can summarise the etiology of atherosclerosis in tabular form

Table has been added into manuscript. Table 1: Risk factors for atherosclerosis identified in the Sub-Saharan African region

4.  Please include the advantages and disadvantages of different imaging techniques in detail

In response to the reviewer’s suggestion. As indicated in line 255-260 the advantages of MRI and disadvantages in accordance with imaging of atherosclerotic plaque.

5. Include more relevant studies in the preclinical evidence

The preclinical evidence stated in the review are current relevant studies which met the reviews topic standards.

6. A summary image of all preclinical studies would improve the quality of the manuscript 

A table is added in place of image summary.

7. Include the table for preclinical studies for a better understanding along with limitations 

Table 3: Summary of the preclinical evidence with the mechanism of radiolabeling, uptake and their limitation.

Reviewer 2 Report

Comments and Suggestions for Authors

Overall, the reviewer agreed with this review. However, some points to be improved were seen in the presetn version as below. 

Introduction gave an imformation on fundamental findings in terms of mechanism.But the same numbering of headlines were seen such as "1. Fibrous Plague, 1. Plaque calcification , 1. Plaque rupture and thrombus formation, ...". On the other hand, no numbering in other sections was seen.

The reviewer expected the results that liposome-based drug delivery system observed or controlled the pathological mechanism of atherosclerosis as seen in the sentences of “atherosclerosis is often diagnosed only … in the early phase.” in Abstract section. However, the story of this review is different from this sentences. The explanatory description of PET might be needed?

Also, the content introduced after the section of Nanoliposome technology did not tell us the solution to the above problem. For examples, in the section of Preclinical Evidence, the concrete information regarding the half-time of liposome over its circulation will be helpful to understand the content concerning the improved imaging duration using liposome-based PET. 

Author Response

Reviewer 2

Overall, the reviewer agreed with this review. However, some points to be improved were seen in the present version as below. 

Introduction gave an information on fundamental findings in terms of mechanism. But the same numbering of headlines was seen such as "1. Fibrous Plague, 1. Plaque calcification, 1. Plaque rupture and thrombus formation, ...". On the other hand, no numbering in other sections was seen.

Tracked changes have been highlighted in red.

The reviewer expected the results that liposome-based drug delivery system observed or controlled the pathological mechanism of atherosclerosis as seen in the sentences of “atherosclerosis is often diagnosed only … in the early phase.” in Abstract section. However, the story of this review is different from this sentence. The explanatory description of PET might be needed?

In response to the comment, the statement “the clinical problems of atherosclerosis mainly involve the difficulty in confirming the plaques or identifying the stability of the plaques in the early phase.”, which is then addressed in the review at line 49-57. Which states: The most common strategy in the treatment or management of atherosclerosis is prophylactic drug therapy. On a clinical level, treating atherosclerosis is primarily focused on relieving CVD symptoms and preventing future cardiovascular events. This can be attributed to the fact that existing clinical screening methods for the diagnosis of atherosclerosis does not provide adequate information on the possible prognosis of the disease state before the first clinical presentation. Plaque burden can be determined by plaque size, histology, chemical composition and bioactivity. Early detection presents new opportunities for primary prevention through lifestyle changes or even drug treatment, especially in patients with high cardiovascular risk.”

Also, the content introduced after the section of Nanoliposome technology did not tell us the solution to the above problem. For examples, in the section of Preclinical Evidence, the concrete information regarding the half-time of liposome over its circulation will be helpful to understand the content concerning the improved imaging duration using liposome-based PET. 

In response to the reviewer, as indicated in lines 402-415, which states: “By combining the diagnosis of atherosclerosis with nanoliposomes, biodistribution can be significantly enhanced, leading to better clinical information on the severity of the disease, better uptake into the plaque, and improved blood clearance. Research has demonstrated that long circulating nanoliposomes radiolabelled with long-lived radionuclides can be injected intravenously and accumulate in atherosclerotic lesions in both cardiovascular disease humans and animal models with increased permeability. PET imaging of radiolabelled nanoliposomes can nonetheless offer a quantitative assessment of biodistribution in vivo, even though its limited spatial resolution makes it difficult to see detailed plaque formation. As macrophages destabilize the atherosclerotic lesion, liposomes are well known for phagocytizing particles that might accumulate in susceptible plaques. Liposomes have demonstrated quick blood clearance and strong macrophage absorption. Rapid blood clearance lowers the high background caused by elevated blood radioactivity for atherosclerotic plaque diagnostic imaging”

Reviewer 3 Report

Comments and Suggestions for Authors

This review discusses the challenges in diagnosing atherosclerosis and the development of radiopharmaceuticals for molecular imaging of atherosclerosis. The manuscript is overall well-presented, however, the subsections of “Imaging Atherosclerosis” should be revised, providing more detail on the existing technologies, radionuclides and liposome strategies. In the current form, it provides only general information.

Specific comments that should be taken into consideration are the following:

1.      In section “Pathophysiology and Aetiology of Atherosclerosis” the numbering of the subsections in italic should be corrected.

2.      The subsection title “ Fibrous Plague” should be corrected to “Fibrous Plaque”

3.      The subsection on nanoliposome technology should be analyzed more specifically related to the various modes of functionalization and radiolabeling as depicted in figures 3 and 4.

4.      In the subsection on Preclinical Evidence of radiolabeled liposomes, the examples of 89Zr-liposomes and 111In-labeled DSPG liposomes are not described in terms of structural elements and mode of interaction with atherosclerosis. Please revise.

5.      In page 11 it is stated that “To achieve delivery with high specificity, liposomes need to be conjugated with functional group molecules such as proteins, antibodies, antibody fragments, carbohydrates, and other small molecules.”, however no such examples are described in the following paragraphs. Please revise.

6.      The abbreviations used in the manuscript should be explained either directly in the text or by adding an abbreviation section.

7.      It was noted that many of the figures in the manuscript are not original fig. 1, fig.4, fig. 5, fig. 6. Only figures 2 and 3 are original. This should be more balanced. Also figure 5 has low resolution and the quality is not optimal.

8.      The references are cited either numerically or by author name/year. Please revise to cite according to the journal’s specifications.

Author Response

Reviewer 3

This review discusses the challenges in diagnosing atherosclerosis and the development of radiopharmaceuticals for molecular imaging of atherosclerosis. The manuscript is overall well-presented, however, the subsections of “Imaging Atherosclerosis” should be revised, providing more detail on the existing technologies, radionuclides and liposome strategies. In the current form, it provides only general information.

Specific comments that should be taken into consideration are the following:

1.      In section “Pathophysiology and Etiology of Atherosclerosis” the numbering of the subsections in italic should be corrected.

Tracked changes have been highlighted in red.

2.      The subsection title “Fibrous Plague” should be corrected to “Fibrous Plaque”

Tracked changes have been highlighted in red.

3.      The subsection on nanoliposome technology should be analyzed more specifically related to the various modes of functionalization and radiolabeling as depicted in figures 3 and 4.

Section highlighted in green lines 338-348 explains the types of nanoparticle labelling addressed in the figures 3 and 4. 

4.      In the subsection on Preclinical Evidence of radiolabeled liposomes, the examples of 89Zr-liposomes and 111In-labeled DSPG liposomes are not described in terms of structural elements and mode of interaction with atherosclerosis. Please revise.

In response to the reviewer’s observation, the mode of interaction for which is states “89Zr-liposomes: There are two possible mechanisms in which nano-liposomes accumulate in plaque macrophages. First, they can migrate into the plaque through splenic or circulating monocytes or by extravasating due to the increased permeability of the blood vessel wall, which results in long-circulating nanoparticles to accumulate within the subendothelial space and eventually phagocytosis by plaque macrophages. Since the Zr-89 is a long-lived isotope, it provides the appropriate amount of time for the long circulating nanoparticle to accumulate in plaque site.”

With regards to the, the chemical name of DSPG is 1,2-Distearoyl sn-glycero-3-phosphoglycerol. The mode of interaction, is stated as “111In-labelled DSPG liposomes is a biocompatible component of liposomes which has shown rapid blood clearance and specific accumulation in macrophage-rich organs.”

5.      In page 11 it is stated that “To achieve delivery with high specificity, liposomes need to be conjugated with functional group molecules such as proteins, antibodies, antibody fragments, carbohydrates, and other small molecules.”, however no such examples are described in the following paragraphs. Please revise.

The statement has now been rephrased to meet the examples stated in the following paragraphs. It now states “To achieve delivery with high specificity, liposomes need to be conjugated with specific linkers and chelators which can encapsulate different forms of radiopharmaceuticals.”

6.      The abbreviations used in the manuscript should be explained either directly in the text or by adding an abbreviation section.

The abbreviation is highlighted in yellow and explained directly in the text.

7.      It was noted that many of the figures in the manuscript are not original fig. 1, fig.4, fig. 5, fig. 6. Only figures 2 and 3 are original. This should be more balanced. Also figure 5 has low resolution and the quality is not optimal.

Figure 5 has been replaced with Table 3 and figure 6 has been changed to an original.

8.      The references are cited either numerically or by author name/year. Please revise to cite according to the journal’s specifications.

The references have been cited numerically.

Round 2

Reviewer 1 Report

Comments and Suggestions for Authors

The authors have improved the manuscript significantly, now it can be accepted in its present form

Author Response

We would like to thank the reviewer for their comment.

Reviewer 3 Report

Comments and Suggestions for Authors

This review discusses the challenges in diagnosing atherosclerosis and the development of radiopharmaceuticals for molecular imaging of atherosclerosis. The revised manuscript is overall well-presented, and the tables added are helpful. 

Specific comments that should be taken into consideration are the following:

1. In line 267, please move citation [46] to its correct place. 

2. Regarding the method of radiolabeling, the authors could include in table 3 or in the respective text, the labeling with desferroxamine for Zr, and via In-oxine converting to In-nitrilotriacetic acid or In-DTPA respectively.

Author Response

Reviewers’ comments

Response

1. In line 267, please move citation [46] to its correct place.

Tracking changing has highlighted change in red at line 266

2. Regarding the method of radiolabelling, the authors could include in table
3 or in the respective text, the labelling with desferroxamine for Zr, and via
In-oxine converting to In-nitrilotriacetic acid or In-DTPA respectively.

In responses to reviewer’s suggestion, line 416-425 states: “According to Lamichhane, et al. (2018) [88] using radiolabelled nanoparticles for molecular imaging also aims to improve delivery, monitoring in vivo pharmacokinetics, and enable well-controlled release. High integration efficiency and adequate retention of radiolabelled drugs are necessary for the use of radiolabelled liposomes in clinical settings. These liposomal compositions can improve uptake or get around the drawbacks of traditional treatments. Because of their adaptability in surface functionalization, nanoparticles offer chances to improve target specificity and label them with different isotopes, which makes them useful as contrast agents. Several labelling techniques can be used to radiolabel liposomes, although doing so may change the pharmacokinetics of the radiolabelled radiopharmaceutical liposome compared to radiopharmaceutical.”